# Health-related quality of life among cervical cancer survivors at a tertiary hospital in Ghana

Kwabena Amo-Antwi[1]*, Ramatu Agambire[2], Thomas O. Konney[1], Samuel B. Nguah[1], Edward T. Dassah[1], Yvonne Nartey[3], Adu Appiah-Kubi[4], Augustine Tawiah[5], Elliot K. Tannor[1], Amponsah Peprah[1], Mavis Bobie Ansah[5], Daniel Sam[5], Patrick K. Akakpo[6], Frank Ankobea[1], Rex M. Djokoto[1], Maame Y. K. Idun[5], Henry S. Opare-Addo[1], Baafour K. Opoku[1], Alexander T. Odoi[1], Carolyn Johnston[7]

1 School of Medicine and Dentistry, Kwame Nkrumah University of Science and Technology, Kumasi, Ghana, 2 Department of Nursing, Garden City University College, Kumasi, Ghana, 3 Division of Epidemiology and Public Health, University of Nottingham, Nottingham, United Kingdom, 4 School of Medical Sciences, University of Health and Allied Sciences, Ho, Ghana, 5 Komfo Anokye Teaching Hospital, Kumasi, Ghana, 6 School of Medical Sciences, University of Cape Coast, Cape Coast, Ghana, 7 University of Michigan, Ann Arbor, Michigan, United States of America

* amoantwikwabena@yahoo.com, kwabena.amo-antwi@knust.edu.gh

**Data Availability Statement:** All relevant data are within the manuscript and its Supporting information files.

## Abstract

### Introduction

Cervical cancer is the second most common female cancer in Ghana. The disease and its treatment significantly affect survivors' health-related quality of life (HRQoL). We determined the overall quality of life (QoL) and identified its predictors among cervical cancer survivors after treatment.

### Materials and methods

A hospital-based cross-sectional analytical study was conducted on 153 disease-free cervical cancer survivors who completed curative treatment between January 2004 and December 2018 at Komfo Anokye Teaching Hospital (KATH) in Kumasi, Ghana. We used the European Organization for Research and Treatment of Cancer core-30 item (EORTC QLQ-C30) and cervical cancer module (EORTC QLQ-CX24) to assess the survivors' overall QoL. QoL domain scores were dichotomised as affected or unaffected by disease and its treatment. Significant differences between the affected and unaffected groups within each QoL domain were determined using the student T-test. We used Kruskal-Wallis and Dunn's tests to examine the difference in QoL domains between treatment types, with significance based on Bonferroni corrections. Multivariable logistic regression was performed to identify predictors of overall QoL. A p-value of less than 0.05 was considered statistically significant.

### Results

One hundred and fifty-three (153) women having a mean age of 58.3 (SD 11.4) years were studied. The overall QoL score was 79.6 (SD 16.0), and 74.5% of survivors reported good

**Funding:** The author(s) received no specific funding for this work. The funders had no role in study design, data collection and analysis, decision to publish, or preparation of the manuscript.

**Competing interests:** The authors have declared that no competing interests exist.

QoL score within the median follow up time of 41.8 months (interquartile range [IQR], 25.5–71.1 months) after cervical cancer diagnosis. Although the majority (66.0–84.3%) of the QoL functioning scale were unaffected, about a fifth (22.2%) to a third (34.5%) of the subjects had perceptual impairment in cognitive and role functioning. Financial difficulties, peripheral neuropathy and pain were most common symptoms reported as affected. A third of the survivors were worried that sex would be painful, and 36.6% indicated that their sexual activity as affected. The overall QoL scores for survivors who had surgery, chemoradiation and radiation-alone were 86.1 (SD 9.7), 76.9 (SD 17.7), and 80.7 (SD 14.7), respectively (p = 0.025). The predictors of survivor's overall QoL were loss of appetite [Adjusted Odd Ratio (AOR) = 9.34, 95% Confidence Interval (CI) = 2.13–35.8, p = 0.001], pain (AOR = 3.53, 95% CI = 1.25–9.31, p = 0.017) and body image (AOR = 5.89, 95% CI = 1.80–19.27, p = 0.003).

## Conclusion

About 75% of the survivors had a good overall quality of life. Primary surgical treatment affords the best prospects for quality of life with the least symptom complaints and financial burden. Loss of appetite, pain or diminution in body image perception predicted the overall quality of life of cervical cancer survivors after treatment.

## Introduction

The distribution of cervical cancer burden is uneven globally, with over 90% of the highest incidence rates of cervical cancer occurring in sub-Saharan Africa [1]. Cervical cancer is the second most common female cancer in Ghana [2]. Although Ghana has significant challenges in the cervical cancer control programme, the logistics and human resource capacity for managing gynaecologic oncologic cases, including cervical cancer, has witnessed tremendous change in the last decade [3, 4]. Increasingly, women with cervical cancer live longer after cancer treatment and are exposed to late side effects of cancer treatment. They often experience symptoms that may adversely affect their health-related quality of life (HRQoL). HRQoL encompasses domains of life that are directly affected by the presence of disease or its treatments [5]. Pain is a common symptom after treatment that may affect their physical activity or emotional well-being [6]. Physical functioning deficit may be as severe as a patient being bedridden or requiring help in performing activities of daily living [5]. While some survivors may exhibit impairment in the ability to execute household chores or job, and other may show deficit in the various elements (tension, worry, irritability and depression) of emotional functioning [3]. Peripheral neuropathy is common in survivors who receive concurrent chemotherapy [7]. Others have recounted their experience with progressive menopausal symptoms in the years following treatment [8, 9]. It is not uncommon for these women to also report perceptual difficulty in memory recall or inability to concentrate on daily tasks. Among young survivors, loss of reproductive organs and external scarring of the genitalia via radical surgery and radiotherapy, respectively, have been reported negatively impacting survivors' psychophysical identity [10]. The inability to reproduce traditionally, although often overlooked, is a significant source of stress or discomfort to the young woman living with or surviving after cancer [11]. Women surviving cervical cancer after radiotherapy reported more sexual problems in their relationship [12]. Several working women may suspend work for a period before and after treatment [3]. Short vagina and chronic lower extremity lymphoedema can occur as late

complications in the surgical patient [13, 14]. The complex interaction of these complaints and symptoms necessitates the evaluation of the quality of life of women surviving cervical cancer.

The most widely used and generally accepted tool for assessing HRQoL in oncology is the European Organization for Research and Treatment of Cancer core-30 item (EORTC QLQ-C30) and cervical cancer module (EORTC QLQ-CX24) questionnaires [15]. These are multidimensional, patient-reported tools that assess aspects of life directly affected by changes in health. Such patient-reported outcomes are novel in the management of cancer in resource-limited settings. Data on HRQoL among cervical cancer survivors are sparse as most studies in Ghana and the sub-region have focused mainly on the epidemiology of cervical cancer [16–19]. QoL domain assessment is ever more critical in this setting, where most women present late with advanced cancer [17, 20]. In Ghana, like in many low and middle-income countries (LMIC), treatment outcomes are often compromised due to limited treatment options, poor supportive care, and suboptimal patient navigation strategies during and after treatment. We, therefore, set out to determine the overall quality of life and identify its predictors among cervical cancer survivors treated at a tertiary hospital in Ghana.

## Material and methods

### Study design

A hospital-based analytical cross-sectional study was conducted from 1st July to 30th September 2019 among women who completed curative treatment for cervical cancer between January 2004 and December 2018 at KATH.

### Setting

The study was conducted at KATH, one of the two largest public cancer treatment centres in Ghana. The hospital is located in Kumasi, the capital city of the Ashanti Region. KATH is a 1200 bed capacity hospital providing diagnostic and treatment services for up to 300,000 patients per year. The Gynaecologic Oncology Unit and Department of Radiation Oncology provide services for over 400 women with genital cancers annually.

Each cervical cancer patient is given an individualized schedule for pre-treatment assessment, definitive treatment, and post-treatment surveillance. The staging of cervical cancer is clinical, with veritable inputs from ancillary procedures and investigations, according to the International Federation of Gynaecologists and Obstetricians (FIGO) staging system of cervical cancer [21]. Haematology, serum biochemistry, ultrasonography (USG), histopathology, cystoscopy, sigmoidoscopy, intravenous urography (IVU), computerized tomography (CT) and magnetic resonance imaging (MRI) assessments form the core of investigations that are done in women presenting with signs and symptoms suggestive of cervical cancer. This evaluation process also seeks to stabilize existing chronic conditions like anaemia, hypertension, and diabetes. Image-guided drainage of hydrometra and pyometra, insertion of nephrostomy tubes and the application of ancillary surgical and medical protocols are also undertaken when necessary. A stage I disease has cancer confined to the cervix, as opposed to FIGO stages II to IV, where the disease extends beyond the cervix.

The mainstay of treatment for women with locally advanced disease (FIGO Stage IB3-IVA) is concurrent chemoradiation [21]. Primary radiotherapy is also indicated in women with early diseases, but are not good candidates (morbidly obese women with attendant detrimental anaesthetic risk, uncontrolled hypertension, or diabetes mellitus) for radical surgical treatment.

The recommended treatment for women with FIGO stage 1A2, 1B1, 1B2 and 2A1 disease is radical hysterectomy and pelvic lymph node dissection, with or without radiotherapy. Post-operative adjuvant radiotherapy (PORT) is considered after surgery, when histopathologic

findings suggest lymph node involvement, positive parametrial or surgical margins among other factors.

A primary radiotherapy section consists of external beam radiation and brachytherapy treatments. The external beam radiation treatment (EBRT) uses a Cirus cobalt-60 teletherapy machine and lately a linear accelerator via box field or parallel-opposed anterior and posterior fields (AP/PA) and two lateral fields. In patients with no disease in the lower third of the vagina, the superior, lateral, and inferior borders of the AP-PA fields are the fourth and fifth lumbar vertebra interspace, 2cm lateral to the pelvic brim on both sides, and 3 cm below the inferior extent of the tumour, respectively. The superior and inferior borders of the lateral fields coincide with that of the AP-PA fields. The anterior and posterior borders of the lateral fields are set with a vertical line anterior to the symphysis and posterior to the entire sacrum, respectively. For diseases extending to the lower third of the vagina, the lateral fields are extended to include the femoral heads, thus affording irradiation of most of the inguinal lymph nodes. The standard hyper-fractionated EBRT of 1.8 to 2.0 Gy are delivered five days a week over 4.5 to 5 weeks, for a total dose of 45 to 50 Gy is given as treatment in women with FIGO stage IB3 to IVA diseases. Cisplatin is given intravenously once a week at a dose of 40 mg per square meter of body-surface area (BSA), with the total dose not to exceeding 70 mg per week. Maximum of six doses of cisplatin are given. In carefully selected cases (e.g., old patients), a hypofractionated EBRT regimen (e.g., 4Gy in 8 biweekly fractions with the same curative intent) is given. The low dose brachytherapy (LDBT) is given after the completion of the EBRT. Our LDBT facility uses a caesium-132 source-the Manchester Dosimetry system with a fletcher semi vaginal applicator. The dose to point A (a reference location 2 cm lateral and 2 cm superior to the cervical os) is 30 to 40 Gy, for a cumulative dose of 75 to 85 Gy, and the cumulative dose to point B (the pelvic wall) is 46 to 50 Gy. In some situations where delay in LDBT is anticipated (backlog of patients awaiting LDBT), the ERBT booster becomes a valuable addition to the prescribed treatment regimen. In times of equipment failure, commonly due to the LDBT machine malfunction, patients are referred to the Radiotherapy Centre in Korle-Bu, Accra, the capital city of Ghana, for High Dose Brachytherapy (HDBT).

All patients are reminded of their appointments a day or two before the due dates. A patient who cannot honour a scheduled visit is often counselled on compliance and given a new date. The patient is labelled as "defaulted treatment" if she failed to honour two or more scheduled visits during pre-treatment assessment or definitive treatment. The first and second review visits are scheduled at two and four weeks, respectively, after treatment. Subsequently, the patient is seen at three-month intervals for the first year and then every 6 months for the next few years. Treatment response is assessed either with World Health Organisation (WHO) response or response evaluation criteria in solid tumours (RECIST) criteria (choice based on the patient's ability to afford CT or MRI scan during pre-treatment workup and post-treatment follow-up) [22].

Cancer survivorship begins at the moment of diagnosis and continues through the cancer trajectories [10]. Women who had survived cervical cancer for five years and beyond after diagnosis were defined as long-term survivors as opposed to short-term survivors who had lived for a period less than five years following a cancer diagnosis.

## Recruitment of study participants

The study population were women who completed curative treatment for cervical cancer between January 2004 and December 2018 at the Gynaecologic Oncology Unit and the Department of Radiation Oncology, KATH (Fig 1). Survivors were eligible if they were disease-free at least six (6) months prior to the study. Women treated for malignancy of other

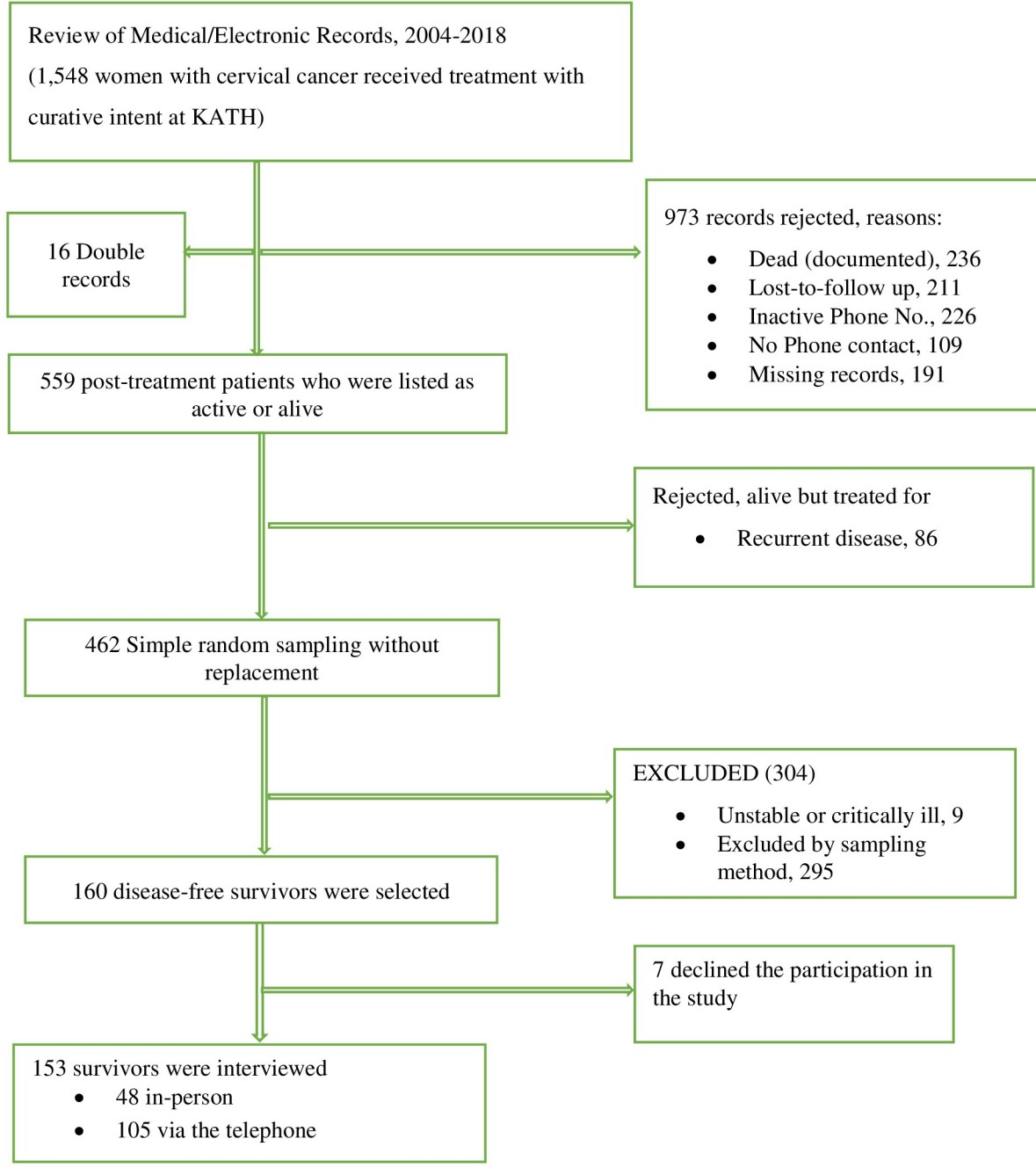

**Fig 1. Flow chart of the sources of and selection of participants into the study.**

anatomic sites, or those critically ill were excluded from the study. The records of 462 women were identified as alive or active based on the surveillance records. The participants for the study were selected using a computer-generated sequence.

## Study variables

The overall QoL score was the dependent variable. The independent variables were the socio-demographic characteristics such as age, level of education, occupation, marital status,

residence, proximity to facility and tribe. Other independent variables studied were the clinical characteristics, including body mass index (BMI), parity, FIGO stage, primary treatment type and the presence of comorbid conditions and time since completion of treatment.

## Data collection

Data were collected by two trained oncology nurses working within the oncology clinic. A two-day training focused on the contents of the questionnaire, identifying subjects based on the inclusion/exclusion criteria, and obtaining consent. After obtaining informed consent, telephone or face-to-face interviews were conducted in Twi, a local language, or English to administer the HRQoL questionnaires to eligible subjects. Telephone interviews were conducted for patients outside Kumasi city or those not scheduled for review within the study period. Data collected were checked for completeness by the principal investigator daily. The Oncology and Gynaecology departments' paper-based and electronic medical records were then reviewed for data on selected survivors. The available information for the subjects was extracted onto a standard data collection sheet. Accuracy and avoidance of data duplication were ensured by linking the following variables between departments: unique identification (ID), age, telephone contact and histology report number. The participants' demographic and clinical characteristics were then extracted, and the data stripped of all identifiers.

## Data collection tools

The survey instruments were the EORTC QoL questionnaires, the generic (EORTC QLQ-C30) and the cervical cancer-specific (EORTC QLQ-CX24) scales, and permission to use the instruments was obtained from the EORTC QoL Group. The 30-item EORTC QLQ-C30 scale comprised global health status/overall QoL subscale; five functional domains (physical, role, cognitive, emotional, and social functioning); three multi-item symptom scales (fatigue, pain, and nausea/vomiting), and six single items that assess additional symptoms commonly reported by cancer patients (dyspnoea, appetite loss, sleep disturbance, constipation, and diarrhoea) and perceived financial difficulties. The EORTC QLQ-CX24 is a 24-item scale, grouped into three (3) multi-item scales, eleven (11) items with symptom experience domain, three (3) items with body image domain, and four (4) items with sexual/vaginal functioning domain. Further, it has single-item scales that assess lymphoedema, peripheral neuropathy, menopausal symptom, sexual worry, sexual activity, and sexual enjoyment. Data on the use of EORTC QLQ-C30 and EORTC QLQ-CX24 scales in Ghana is limited but have been validated for use in other parts of sub-Saharan Africa [23]. The researchers adopted the closest locally relevant interpretation of questions. All scores on the EORTC QLQ-C30 and QLQ-CX24 were transformed into 0 to 100 scale, according to the EORTC QLQ scoring manual [24]. A higher score on the global health/overall QoL and functioning scale represented a better level of functioning. A higher score on the symptom scale represented worse level of symptoms or problems.

## Sample size

We assumed that our study's overall QoL score was similar to that observed by Khalil et al [25]. in their assessment of the impact of cervical cancer on QoL: beyond the short term. Using a standard deviation [SD] of 21.5 and desiring a margin of error of 3.5 percentage points from the actual population overall QoL score, an estimated sample size of 148 had adequate power to detect the sample's overall QoL score. Allowing 10% for contingency, inappropriate and nonresponses, our estimated sample size was 160.

## Statistical analysis

The EORTC QLQ-C30 and QLQ-CX24 scores, demographic, clinical and treatment-related variables were summarised using tables, proportions, means and SD, and medians with inter-quartile ranges (IQR). The EORTC QLQ-C30 and QLQ-CX24 domain scores were later dichotomised; individuals with a functional domain or overall QoL score below 75 were considered affected by disease or treatment (75 or more indicated "good functioning"), whereas, on the symptom scale, a score of 25 and above were deemed affected or problematic (below 25 indicated "good functioning") [26]. Significant differences between the affected and unaffected groups within each QoL domain were determined using the student T-test. Normality tests were carried out for the overall QoL, functioning and symptom scores. We used Kruskal-Wallis and Dunn's tests to examine the difference in QOL domains between treatment types, with significance based on Bonferroni corrections. Stepwise logistic regression was performed to identify predictors of overall QoL (included only variables with p < 0.25) [26]. A multivariable logistic regression analysis, including only clinically significant variables, was then performed. All tests were two-sided tests; p < 0.05 was considered statistically significant. All analyses were performed using Stata 13 (StataCorp, College Station, TX, USA).

## Ethics

The research protocol, questionnaire, consent statement were approved by the Committee on Human Research, Publications and Ethics (CHRPE), Kwame Nkrumah University of Science and Technology and KATH (CHRPE/AP/661/19). Since most of the interviews were conducted over the telephone, informed verbal consent was obtained from all respondents before the interview, similar to the method used by Karasik et al. [27] Respondents were informed in Twi or English about their rights of voluntary participation and withdrawal from the study. They were aware that deciding not to participate in the study would not affect their post-treatment care. The anonymity of the data was assured.

## Results

### Sociodemographic and clinical characteristics of cervical cancer survivors

One hundred and fifty-three (153) consenting women were studied (Fig 1). Data for all the participants were complete. The mean age of the survivors was 58.3 (SD 11.4) years, with a median follow up time of 41.8 months (interquartile range [IQR], 25.5 to 71.1 months) after cervical cancer diagnosis (Table 1). About a third (34.6%) of the women were long-term survivors. About 40% had no formal education, and over two-thirds (68.6%) had informal occupations. About 57% were anaemic at the time of treatment, necessitating haemotransfusion in nearly a third (32.0%) of the survivors (Table 2). Most participants (37.3%) were treated for FIGO stage III cervical cancer. Primary chemoradiation was the treatment modality of choice in nearly half (47.7%) of the women. Twenty-eight (18.3%) participants underwent primary surgical treatment with or without adjuvant radiotherapy.

### EORTC QLQ—C30 & CX24 scale scores among the cervical cancer survivors

The Overall QoL score was 79.6 (SD 16.2), with 74.5% reporting good overall QoL (Table 3). Although the majority (66.0–84.3%) of the QoL domains on the EORTC QLQ-C30 functioning scale were unaffected, about a fifth (22.2%) to third (34.5%) of the subject had perceptual impairment in cognitive and role functioning. On the EORTC QLQ-C30 symptom scale, 32.0% reported pain as problematic, and 47.7% had financial difficulties. About a fifth (20.9%)

**Table 1. Sociodemographic characteristics of cervical cancer survivors after treatment.**

| Characteristics | Number[a] (N = 153) | Percentage (%) |
|---|---|---|
| **Age at interview, years** | | |
| <50 | 36 | 23.5 |
| 50–59 | 48 | 31.4 |
| 60–69 | 40 | 26.1 |
| > = 70 | 29 | 19.0 |
| Mean ± SD | 58.3±11.4 | |
| **Follow up time (years)** | | |
| < 5 | 100 | 65.4 |
| ≥ 5 | 53 | 34.6 |
| Median (IQR), months | 41.8 (25.5–71.1) | |
| **Education** | | |
| No formal education | 61 | 39.9 |
| Basic education | 75 | 49.0 |
| Secondary & Tertiary education | 17 | 11.1 |
| **Employment** | | |
| Informal | 105 | 68.6 |
| Formal | 16 | 10.5 |
| Unemployed | 32 | 20.9 |
| **Marital status** | | |
| Married/Co-habitating | 74 | 48.4 |
| Single | 79 | 51.6 |
| **Town of residence** | | |
| Urban | 99 | 64.7 |
| Rural | 54 | 35.3 |
| **Region of residence** | | |
| Western Region | 10 | 6.5 |
| Ashanti Region | 81 | 52.9 |
| Brong Ahafo Region | 28 | 18.3 |
| Central Region | 10 | 6.5 |
| Other Regions [b] | 24 | 15.6 |
| **Tribe/ethnicity** | | |
| Akan | 123 | 80.4 |
| Mole Dagbane | 11 | 7.2 |
| Others | 19 | 12.4 |

*Note*: IQR: interquartile range; SD: Standard deviation.

[a] Values are given as number unless otherwise stated; Other Regions

[b]: Greater Accra, Eastern, Upper East and West, Volta and Northern Regions of Ghana

of the survivors perceived their sleep to be affected. Additionally, on the cervical cancer-specific scale, ERTOC QLQ-CX24, 76.6% of the survivors reported good functioning with regards to body image, while 37.2% had a problem with peripheral neuropathy. Furthermore, a third (30.1%) of the survivors were worried that sex would be painful. Over a third indicated that their sexual activity was problematic.

A visible change in overall QOL and functioning occurred after ten (10) years, while the perception of financial difficulties only improved after fifteen (15) years. The course of overall QoL score had an inverse relationship with the pattern of symptom complaints (Figs 2–4)

**Table 2. Clinical characteristics of cervical cancer survivors during treatment.**

| Characteristics | Number[a] N = 153 | Percentage (%) |
|---|---|---|
| **BMI (Kg/m$^2$)** | | |
| Underweight ($<$ 18.5) | 10 | 7.9 |
| Normal (18.5–24.9) | 63 | 49.6 |
| Over-weight (25–29.9) | 35 | 27.6 |
| Obese ($>$ 30) | 19 | 15 |
| Mean ± SD | 24.8±5.1 | |
| **FIGO Stage** | | |
| Stage I | 42 | 27.4 |
| Stage II | 54 | 35.3 |
| Stage III | 58 | 37.3 |
| **Treatment Modality** | | |
| Surgery Alone | 17 | 11.1 |
| Surgery Plus Radiotherapy | 11 | 7.2 |
| Radiotherapy Alone | 52 | 34.0 |
| Chemoradiation | 73 | 47.7 |
| **Comorbidities** | | |
| Anaemia Alone | 51 | 33.3 |
| Anaemia/HTN | 30 | 19.6 |
| Hypertension Alone | 19 | 12.4 |
| Other comorbidities [b] | 11 | 7.2 |
| No comorbidities | 42 | 27.5 |
| **Haemoglobin level (g/dL)** | | |
| Severe Anaemia ($<$ 8) | 6 | 3.9 |
| Moderate Anaemia (8–10.9) | 47 | 30.7 |
| Mild Anaemia (11–11.9) | 34 | 22.2 |
| Normal ($\geq$ 12) | 66 | 43.1 |
| Mean ± SD | 11.4 ±1.6 | |
| **Haemotransfusion** | | |
| No | 111 | 72.6 |
| Yes | 42 | 27.4 |
| **Units of Blood transfused, (N = 42)** | | |
| $\leq$ 2 | 18 | 42.9 |
| 3–4 | 24 | 57.1 |
| Mean ± SD | 3.1±1.6 | |

[a] Values are given as number unless otherwise stated; Other comorbidities

[b]: Deep vein thrombosis, diabetes, human immunodeficiency, cholelithiasis, haemorrhoid, left ventricular hypertrophy and paraparesis

## EORTC QLQ-C30 and QLQ-CX24 domains and primary treatment modality

Kruskal-Wallis test revealed significant differences in overall QoL (p = 0.027), financial difficulties (p = 0.019), body image (p = 0.039), and symptoms experienced (p = 0.043), with regards to the treatment received by the cervical cancer survivors (primary surgical treatment, n = 28; radiation alone, n = 52; chemoradiation, n = 73). A primary surgical treatment afforded a better global health status and body image and fewer symptoms (Table 4). Dunn's pairwise

**Table 3. EORTC QLQ-C30 & QLQ-CX24 of among cervical cancer survivors after treatment.**

| Variables [a] | | | | Scoring | |
|---|---|---|---|---|---|
| | Number of Items | Mean (SD) | 95% C.I. | < 25 (%) | ≥ 75 (%) |
| **QLQ-C30 Functional scale [b]** | | | | | |
| Overall QOL (GHS) score | 2 | 79.6 (16.0) | 77.1–82.2 | 25.5 | 74.5 |
| Physical functioning | 5 | 88.2 (14.1) | 86.0–90.5 | 20.9 | 79.1 |
| Role functioning | 2 | 87.1 (19.5) | 83.9–90.2 | 22.2 | 77.8 |
| Cognitive functioning | 2 | 80.5 (24.9) | 76.6–84.5 | **34.0** | 66.0 |
| Emotional functioning | 4 | 87.8 (18.1) | 84.9–90.7 | 21.6 | 78.4 |
| Social functioning | 2 | 91.8 (18.4) | 88.8–94.7 | 15.7 | 84.3 |
| **QLQ-C30 Symptom scale [c]** | | | | | |
| Energy/fatigue | 3 | 9.0 (11.3) | 7.2–10.8 | 91.5 | 8.5 |
| Nausea and vomiting | 2 | 3.3 (10.3) | 1.6–4.9 | 95.4 | 4.6 |
| Pain | 1 | 17.9 (24.0) | 14.0–21.7 | 68.0 | **32.0** |
| Short of breath | 1 | 5.7 (15.2) | 3.2–8.1 | 86.3 | 13.7 |
| Sleep disturbance | 1 | 9.3 (19.7) | 6.2–12.5 | 79.1 | **20.9** |
| Lack of appetite | 1 | 6.5 (18.0) | 3.7–9.4 | 86.9 | 13.1 |
| Constipation | 1 | 8.7 (19.0) | 5.6–11.8 | 80.4 | 19.1 |
| Diarrhoea | 1 | 3.0 (9.6) | 1.5–4.6 | 90.9 | 9.1 |
| Financial difficulty | 1 | 36.2 (42.7) | 29.4–43.0 | 52.3 | **47.7** |
| **QLQ-CX24 Functional scale [b]** | | | | | |
| Body image | 3 | 85.3 (26.0) | 81.2–89.5 | 23.5 | 76.5 |
| Sexual activity | **1** | 77.6 (34.0) | 72.1–83.0 | **36.6** | 63.4 |
| Sexual enjoyment, N = 56 | **1** | 50.0 (40.7) | 39.1–60.9 | **71.4** | 28.6 |
| Sexual/vaginal functioning, N = 56 | **4** | 71.5 (24.2) | 65.0–78.0 | **58.9** | 41.1 |
| **QLQ-C24 Symptom scale [c]** | | | | | |
| Symptom experience | 11 | 10.0 (10.3) | 8.4–11.7 | 91.5 | 8.5 |
| Lymphedema | 1 | 7.0 (19.7) | 3.8–10.1 | 86.9 | 13.1 |
| Peripheral neuropathy | 1 | 17.6 (26.2) | 43.2–51.6 | 62.8 | 37.2 |
| Menopausal symptoms | **1** | 12.4 (25.0) | 8.4–16.4 | 75.8 | 24.2 |
| Sexual worry | 1 | 20.9 (35.4) | 15.3–26.6 | 69.9 | 30.1 |

*Note*: EORTC QLQ–C30: European Organization for Research and Treatment of Cancer core-30 items; EORTC QLQ–CX24 EORTC: Cervical cancer module; SD: standard deviation, C.I.: confidence interval

[a] Variables: Number of Variables, N = 153 unless otherwise indicated;

[b] For the functioning scale, a score below 75 were considered affected by disease or treatment (75 or more indicated "good functioning").

[c] For the symptom scale, a score of 25 and above were deemed affected or problematic (below 25 indicated "good functioning"). A normality test was done using the histogram. Significant differences between the affected and unaffected groups within each QoL domain were determined using the student T-test.

comparison further confirmed a difference in the score for overall QoL (p = 0.0036), financial impact (p = 0.0013), body image (p = 0.0013), and symptom experience (p = 0.0068) regarding surgery and chemoradiation (Table 5). However, women who had surgery differed from radiation-alone treatment based on body image (p = 0.0212), and financial impact (p = 0.0206).

## Predictors of overall QoL among the cervical cancer survivors

From multivariable logistic regression the overall QoL was significantly associated with loss of appetite (AOR = 9.34, 95% CI = 2.36–36.94, p = 0.001), pain (AOR = 3.53, 95% CI = 1.30–9.56, p = 0.016) and body image (AOR = 5.89, 95% CI = 1.80–19.27, p = 0.003) (Table 6).

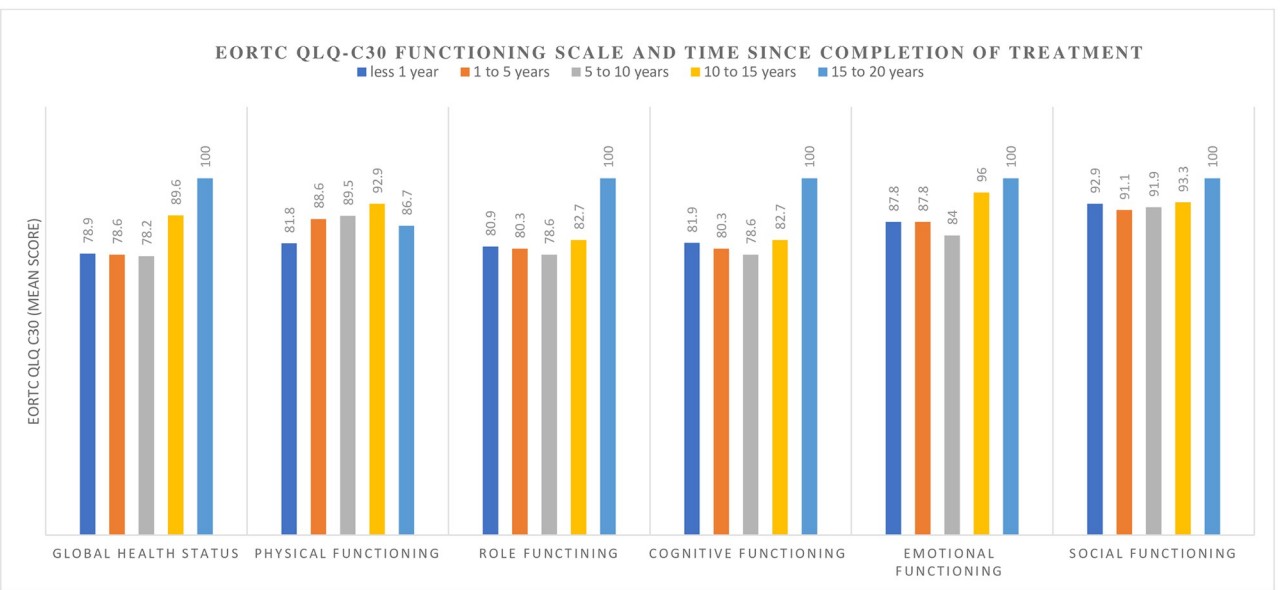

**Fig 2. The change in EORTC QLQ C30 functioning scores of cervical cancer survivors after treatment at Komfo Anokye Teaching Hospital, Kumasi, Ghana.**

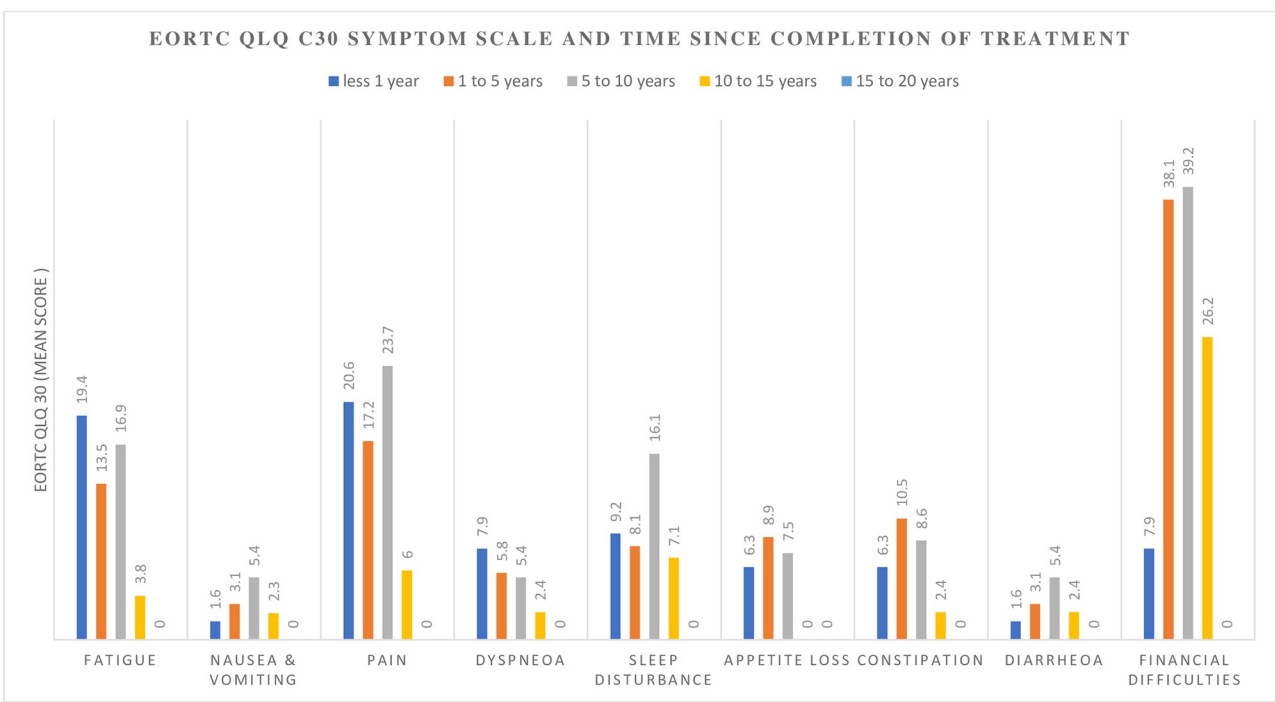

**Fig 3. The change in EORTC QLQ C30 symptoms scores in the cervical cancer survivors after treatment at Komfo Anokye Teaching Hospital, Kumasi, Ghana.**

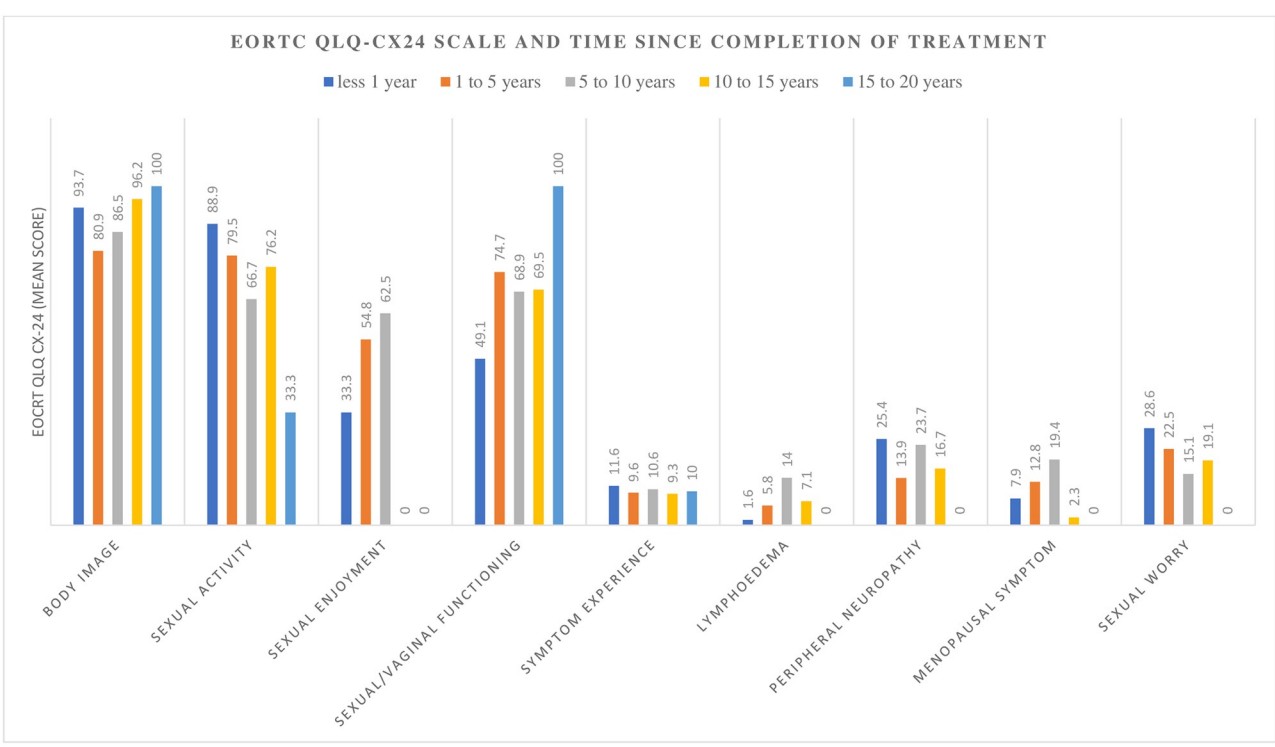

**Fig 4. The change in EORTC QLQ CX-24 functioning and symptoms scores of the cervical cancer survivors after treatment at Komfo Anokye Teaching Hospital, Kumasi, Ghana.**

## Discussion

Investigating quality of life among cervical cancer survivors is novel in Ghana. Although gynaecologic oncologists recognize the quality of life assessment as important, it is often not the central theme in managing cervical cancer in the sub-region. Instead, emphasis is placed on 5-year survival as the major oncologic outcome. HRQoL, which focus on patients' perception of disease and its treatments, provides critical information that cannot be obtained by the conventional clinical and functional measurements.

In high-income countries, the incidence of cervical cancer cases has reduced [20]. Most are diagnosed at an early stage due to the robust screening programmes. On the contrary, Ghana's cervical cancer control programme has a huge resource deficit. Only 5.2% of women diagnosed with cervical cancer had FIGO Stage I disease and were likely to benefit from surgery [28]. In this study, women who had primary surgical treatment had a better overall QoL, consistent with earlier studies [8, 9, 29]. Surgery offered the best prospects for the body image, with the least financial burden and complaints after treatment. Consistently, minimizing the use of postoperative adjuvant radiotherapy maximize the gains from primary surgical treatment. The use of cross-sectional imaging reduces the error associated with clinical FIGO staging and the likelihood of multimodal treatment in patients deemed to have diseases confined to the cervix [21, 30].

We observed a noticeable improvement in the perceived overall QoL over ten to fifteen years after completion of treatment. This is attributable to the perception of reducing symptoms and complaints with increasing periods of survivorship [10]. The recovery in overall QoL occurs earlier in younger subjects and among survivors treated at centres provides optimum

**Table 4. EORTC QLQ C-30 and QLQ CX-24 domains and treatment type received by the survivors.**

| EORTC QLQ | [a] Surgery (+/-RT) | Radiation alone | Chemoradiation | p |
|---|---|---|---|---|
| | Mean (+/-SD) | Mean (+/-SD) | Mean (+/-SD) | |
| **QLQ-C30 Functional scale** | | | | |
| Overall QOL (GHS) score | 86.1 (9.7) | 80.7(15.2) | 76.9 (17.7) | 0.025 |
| Physical function | 91.0 (11.1) | 87.4 (14.1) | 87.7 (15.1) | 0.663 |
| Role functioning | 89.5 (19.0) | 90.1(15.9) | 84.0 (21.7) | 0.297 |
| Cognitive functioning | 83.4 (24.0) | 83.2 (26.6) | 77.6 (26.2) | 0.445 |
| Emotional functioning | 93.2 (15.0) | 88.5 (17.61) | 85.3 (19.6) | 0.079 |
| Social functioning | 97.1 (11.0) | 91.2 (18.5) | 90.2 (20.3) | 0.384 |
| **QLQ-C30 Symptom scales** | | | | |
| Energy/fatigue | 7.5 (11.7) | 9.1 (12.0) | 9.4. (10.7) | 0.977 |
| Nausea and vomiting | 2.4 (6.0) | 3.8 (12.2) | 3.2 (10.3) | 0.992 |
| Pain | 14.9 (19.4) | 14.7 (20.8) | 21.2 (27.4) | 0.495 |
| Short of breath | 2.4 (8.7) | 5.7 (14.3) | 6.8 (17.5) | 0.789 |
| Sleep disturbance | 3.6 (10.5) | 10.9 (20.6) | 10.5 (21.6) | 0.526 |
| Lack of appetite | 3.5 (10.5) | 5.1 (15.3) | 10.5 (22.8) | 0.638 |
| Constipation | 9.5 (20.0) | 9.6 (20.2) | 7.8 (18.0) | 0.928 |
| Diarrhoea | 0 | 3.2 (9.9) | 4.5 (11.4) | 0.570 |
| Financial difficulty | 15.7 (34.0) | 34.0 (38.2) | 45.7 (46.3) | 0.022 |
| **QLQ-CX24 Functional scale** | | | | |
| Body image | 95.7 (15.6) | 87.9 (21.9) | 79.5 (30.1) | 0.039 |
| Sexual activity | 84.5 (30.7) | 79.5 (35.0) | 73.5 (34.2) | 0.21 |
| Sexual enjoyment, n = 56 | 57.1 (46.0) | 31.1(36.7) | 56.9 (39.8) | 0.141 |
| Sexual/vaginal functioning, n = 56 | 80.5 (25.9) | 76.6 (19.8) | 67.7 (25.6) | 0.370 |
| **QLQ-C24 Symptom scales** | | | | |
| Symptom experience | 5.9 (5.7) | 9.4 (10.1) | 12.0 (11.4) | 0.043 |
| Lymphedema | 6.0 (15.9) | 5.7 (15.7) | 8.2 (23.4) | 0.997 |
| Peripheral neuropathy | 9.5 (17.8) | 19.2 (26.7) | 19.6 (28.2) | 0.35 |
| Menopausal symptoms | 9.5 (20.0) | 14.1(25.9) | 12.3 (26.9) | 0.788 |
| Sexual worry | 10.7 (28.8) | 19.6 (36.6) | 27.1 (36.9) | 0.115 |

*Note*: EORTC: European Organisation for Research and Treatment of Cancer generic (QLQ C-30) and cervical cancer-specific (QLQ CX-24) scale domains; QOL: quality of life; GHS: global health status;

[a] Surgery (+/-RT): Radical hysterectomy plus pelvic lymph node dissection +/- postoperative adjuvant radiotherapy.

[b] p: The P-value for the Kruskal Wallis tests, to detect differences in QOL domain score with regards to primary treatment received by the survivor.

supportive care [9]. It is evident that our cervical cancer patients also struggle with physical problems and discomfort for prolonged periods after treatment.

Only a third (36.6%) of our disease-free cervical cancer survivors were sexually active. This proportion is comparatively higher than that reported among survivors in Iran [31]. Expectedly, the more liberal or secular Ghanaian society will report a higher level of sexual activity [32]. We also saw a decreased sexual activity with increasing periods of survivorship even though sexual functioning (feeling that vagina is short, tight, dry, or one will experience dyspareunia) and sexual enjoyment improved. Sexual function, including desire, is often intact in older women, but its course decreases with increasing age [33]. This was expected in a group of women with an average of 58 years. Efforts to enhance overall QoL should also be focused on sexual rehabilitation. This should start as early as possible to allay any anxieties about sex, especially in those that receive radiation treatment.

**Table 5. Dunn's post hoc test of selected QoL domains and primary treatment.**

| QOL domains | Surgery Vs Radiation Alone | Surgery Vs Chemoradiation | Radiation Vs Chemoradiation |
|---|---|---|---|
| **Overall QOL (GHS)** | | | |
| Difference in rank sum | 1.5653 | 2.6891 | 1.2720 |
| Significant p[a] < 0.05? | No (0.0588) | **Yes** (0.0036) | No (0.1017) |
| **Financial impact** | | | |
| Difference in rank sum | -2.0304 | -3.0024 | -1.0551 |
| Significant p[a] < 0.05? | **Yes** (0.0212) | **Yes** (0.0013) | No (0.1457) |
| **Body image** | | | |
| Difference in rank sum | 2.0423 | 3.0160 | 1.0564 |
| Significant p[a] < 0.05? | **Yes** (0.0206) | **Yes** (0.0013) | No (0.1454) |
| **Symptom experience** | | | |
| Difference in rank sum | -1.2494 | -2.4689 | -1.4105 |
| Significant p[a] < 0.05? | No (0.1058) | **Yes** (0.0068) | No (0.0792) |

*Note*: QOL: quality of life; GHS: global health status.

Difference in rank sum: z-test statistics. p[a]: The P-value for Dunn's Post Hoc test, interpreted as "yes" if a significant difference was confirmed.

A third of all working women suspend work before and after treatment for cancer [3]. Close to half of the survivors in this study perceived their financial situation as affected or problematic. Most of the women were employed by the informal sector with high income insecurity and limited access to social benefits through institutionalized schemes [34]. Even though Ghana has National Health Insurance Scheme (NHIS), it is a drain on their finances if the majority of the hospital bills are funded out-of-pocket [35]. The scheme should be expanded to absorb most hospital bills, especially in women undergoing cervical cancer treatment. Financial assistance from families and institutions to re-integrate survivors into former roles may mitigate the survivor's financial plight.

Several studies have published predictive factors for QoL in cervical cancer survivors, ranging from intimate spousal issues, reproductive concerns, and spiritual well-being to social issues. However, in this current study, we observed that the complaints of pain, loss of appetite, and diminution in body image were predictive of overall QoL among our cervical cancer survivors. In an earlier study, 5–10% of cervical cancer survivors complained about pain [6]. Expectedly, about a third of the women surveyed recounted their experience with pain as problematic. The cause of the pain in the survivors may not necessarily be attributable to cancer or its management [6]. Current interventions in cancer care may afford survivors the chance to live longer and experience pain due to osteoarthritis, secondary to the ageing process or senescence [10]. Evaluation of pain and provision of adequate analgesia during posttreatment surveillance will be cardinal in enhancing the QoL of our survivors.

Loss of appetite also had a negative effect on the QoL among cervical cancer survivors in Iran and Bangladesh [8, 31]. Not knowing what to eat and complaining of pain may underline the lack of appetite for regular stables. Early nutritionist support during and after treatment is critical in addressing misconceptions about diet and cancer. This may be key in enhancing survivors' QoL [36].

In most of the cultures in Sub-Saharan Africa, childbearing signifies an expression of femininity [37]. The loss of reproductive organs and external scarring of the genitalia due to radical surgery and radiotherapy, respectively, have been reported to negatively impact survivors' psychophysical identity [11]. It is then expected that the overall QOL would be low for survivors who reported the body image affected. It was also noted that improvement in sexual

**Table 6. Predictors of Overall QoL of the cervical cancer survivors after treatment.**

| Variables | Overall QOL (GHS) score, n (%) | | p-value, OR (95% CI) | |
|---|---|---|---|---|
| | Unaffected | Affected | OR* (95%CI) | OR† (95% CI) |
| **Survivorship, years** | | | **0.562** | |
| ≥ 5 | 38 (33.3) | 15 (38.5) | 1 | |
| < 5 | 76 (66.7) | 24 (61.5) | 0.80 (0.38–1.70) | |
| **Level of education** | | | **0.336** | |
| Formal | 66 (57.9) | 26 (66.7) | 1 | |
| No formal education | 48 (42.1) | 13 (33.3) | 0.69 (0.32–1.47) | |
| **Role functioning** | | | **0.002** | **0.136** |
| Unaffected | 96 (84.2) | 23 (59.0) | 1 | 1 |
| Affected | 18 (15.8) | 15 (41.0) | 3.71 (1.65–8.36) | 0.76 (0.75–6.97) |
| **Fatigue** | | | **0.835** | |
| Unaffected | 104 (91.2) | 36 (92.3) | 1 | |
| Affected | 10 (8.8) | 3 (7.7) | 0.87 (0.22–3.33) | |
| **Pain** | | | **0.0001** | **0.016** |
| Unaffected | 90 (79.0) | 14 (35.9) | 1 | 1 |
| Affected | 24 (21.0) | 25(64.1) | 6.70 (3.02–14.82) | 3.42 (1.25–9.31) |
| **Loss of appetite** | | | **0.0001** | **0.001** |
| Unaffected | 109 (95.6) | 24 (61.5) | 1 | 1 |
| Affected | 5 (4.4) | 15 (38.5) | 13.62 (4.52–41.11) | 8.70 (2.13–35.58) |
| **Constipation** | | | **0.001** | **0.152** |
| Affected | 99 (86.8) | 24 (61.5) | 1 | 1 |
| Unaffected | 15 (13.2) | 15 (38.5) | 4.13 (1.78–9.59) | 2.26 (0.76–7.39) |
| **Diarrhoea** | | | **0.035** | **0.111** |
| Unaffected | 107 (93.9) | 32 (82.0) | 1 | 1 |
| Affected | 7 (6.1) | 7 (18.0) | 3.34 (1.09–10.24) | 0.21 (0.03–1.54) |
| **Financial difficulties** | | | **0.0001** | **0.058** |
| Unaffected | 71 (62.3) | 9 (23.1) | 1 | 1 |
| Affected | 43 (37.7) | 30 (76.9) | 5.50 (2.39–12.69) | 2.94 (0.96–8.98) |
| **Body image** | | | **0.0001** | **0.003** |
| Unaffected | 101 (88.6) | 16 (41.0) | 1 | 1 |
| Affected | 13 (11.4) | 30 (59.0) | 11.17 (4.72–26.41) | 5.89 (1.80–19.27) |
| **Sexual activity** | | | **0.624** | |
| Unaffected | 71 (62.3) | 26 (66.7) | 1 | |
| Affected | 43 (37.7) | 13 (33.3) | 0.82 (0.38–1.78) | |

*Note*: **OR***: Crude odds ratio OR†: Adjusted odds ratio. QOL: quality of life; GHS: global health status.

CI confidence interval. Significant differences between the affected and unaffected groups within each QoL domain were determined using the student T-test

The model did not include sexual functioning and enjoyment because some subjects did not report any recent (within the last 4 weeks of the study) sexual activity.

Stepwise regression: variables (Age, body mass index, parity, tribe, occupation, pre-treatment haemoglobin level, physical functioning, emotional functioning, cognitive functioning, insomnia, symptom experience) with p ≥ 0.25 were not included in the multivariable regression model.

functioning and enjoyment of the survivors occurred during the time of perceived improvement in the body image.

Our study makes it reasonable to hypothesize that cervical cancer survivors with a perception of low body image, those complaining of pain or loss of appetite will have a low overall QoL. Patient-specific supportive care and surveillance plan should extend well into the periods of survivorship.

A major strength of the study is its contribution to the quest of improving cervical cancer care in Ghana. However, the study has some limitations. The EORTC assessment scale has not been validated in Ghana. The researchers adopted the closest locally relevant interpretation of EORTC QLQ item questions. The QoL of cancer survivors' changes over time, hence a cross-sectional design is limited in its ability to detect the pattern of overall QoL across the cancer trajectory. This is a single institution study, and the results may not be generalizable to the entire disease-free cervical cancer survivors in Ghana.

## Conclusion

About 75% of the survivors had a good overall quality of life. Primary surgical treatment affords the best prospects for quality of life with the least symptom complaints and financial burden. Complaints of pain or loss of appetite or reported diminution in body image perception predicted the cervical cancer survivor's overall quality of life treated at our centre. Strategies to improve traditional oncologic outcomes (disease-free, progression-free, and overall survival) should also address self-reported complaints (loss of appetite and pain) and distortions in the perception of the self (body image).

## Supporting information

**S1 File. Contains information on anonymized participants' data.**
(XLSX)

**S2 File. Contains filled questionnaire on equity and inclusivity in research.**
(DOCX)

## Acknowledgments

We would like to extend our sincere gratitude to all staffs of the Unit of Gynaecologic Oncology and the Department of Radiation Oncology of KATH, especially Abigail Osei-Mensah and Abigail Owusu Sekyere for their immense role during the interview. The authors acknowledge the support offered by the following people during the preparation of the manuscript: Dr. Abdul-Razak Abdul-Munin and Dr. Ernest Bawuah Bonsu.

## Author Contributions

**Conceptualization:** Kwabena Amo-Antwi, Thomas O. Konney, Yvonne Nartey, Baafour K. Opoku, Alexander T. Odoi.

**Data curation:** Kwabena Amo-Antwi, Mavis Bobie Ansah.

**Formal analysis:** Kwabena Amo-Antwi, Samuel B. Nguah.

**Methodology:** Kwabena Amo-Antwi, Samuel B. Nguah, Edward T. Dassah, Yvonne Nartey.

**Project administration:** Kwabena Amo-Antwi, Ramatu Agambire, Mavis Bobie Ansah.

**Writing – original draft:** Kwabena Amo-Antwi, Thomas O. Konney, Patrick K. Akakpo.

**Writing – review & editing:** Kwabena Amo-Antwi, Ramatu Agambire, Thomas O. Konney, Edward T. Dassah, Adu Appiah-Kubi, Augustine Tawiah, Elliot K. Tannor, Amponsah Peprah, Daniel Sam, Patrick K. Akakpo, Frank Ankobea, Rex M. Djokoto, Maame Y. K. Idun, Henry S. Opare-Addo, Baafour K. Opoku, Carolyn Johnston.

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
