## [Decision Letter · Decision Letter 0]

23 Feb 2022

PONE-D-21-32659Health-related quality of life among cervical cancer survivors at a tertiary hospital in GhanaPLOS ONE

Dear Kwabena Amo-Antwi,

Thank you for submitting your manuscript to PLOS ONE. After careful consideration, we feel that it has merit but does not fully meet PLOS ONE’s publication criteria as it currently stands. Therefore, we invite you to submit a revised version of the manuscript that addresses the points raised during the review process.

Please read and respond to all of the peer review comments. You should provide a point-by-point response to explain any changes you have (or have not) made to the original article and be as specific as possible in your responses. Before your re-submission, please go through the submission guideline and make sure that the format of the revised manuscript is in accordance with the journal requirement.

We look forward to receiving your revised manuscript.

Kind regards,

Alison Wang

Academic Editor

PLOS ONE

https://journals.plos.org/plosone/s/fileid=ba62/PLOSOne_formatting_sample_title_authors_affiliations.pdf".

**Additional Editor Comments :**

1. Ethical issue: Verbal consent of the participants is a major concern. Do you have any evidence to show that you have gotten the consent from the participants? 

**Comments to the Author**

1. Is the manuscript technically sound, and do the data support the conclusions?

Reviewer #1: Yes

Reviewer #2: No

2. Has the statistical analysis been performed appropriately and rigorously? 

Reviewer #1: No

Reviewer #2: No

3. Have the authors made all data underlying the findings in their manuscript fully available?

Reviewer #1: Yes

Reviewer #2: Yes

4. Is the manuscript presented in an intelligible fashion and written in standard English?

Reviewer #1: Yes

Reviewer #2: No

5. Review Comments to the Author

**Reviewer #1**: Health-related quality of life among cervical cancer survivors at a tertiary hospital in Ghana

The reviewed manuscript titled "Health-related quality of life among cervical cancer survivors at a tertiary hospital in Ghana" is a well-written article. The study presents the results of the original research. The title of the article encourages to read its content carefully. The topic discussed in the article is worth disseminating due to the rarity of this content in the local culture and the opportunity to learn about the quality of life in the course of cervical cancer in a different cultural area and in a low-income country.

Importantly, the researchers obtained the consent of the Committee on Human Research, Publications and Ethics (CHRPE) of the School of Medicine and Dentistry, Kwame Nkrumah University of Science and Technology and KATH (CHRPE / AP / 661/19)to conduct the research. The women gave verbal consent to participate in the study, and for their clinical details to be shared, following the standard approach at KATH.

The structure of the article is clear. The article is presented in an intelligible fashion and is written in standard English.

Research methodology:

The selected research methods are reliable and valid - selected adequately to the topic from the area of quality-of-life questionnaires developed by the EORTC (European Organization for Research and Treatment of Cancer - Quality of Life Group). The sample size is sufficient, and the selection scheme of patients has been carried out correctly and is clearly presented in a graphic form. The applied statistical procedures, interpretation, and presentation do not raise any objections. Experiments, statistics, and other analyses are performed to a high technical standard and are described in sufficient details. However, I would like to mention one thing the authors should do to improve the manuscript, which relates to table 4: the comparison of three groups divided according to the treatment reveceived, where the Kruskal-Wallis test revealed p <0.05, which means that at least one of the studied groups differs from the other group. To find out exactly which groups differ from each other, a „post-hoc test" should be used, e.g. Dunn’s test.

I recommend using a "post-hoc test" for the following variables:

1. Global health status (p = 0.027)

2. Finacial difficulties (p = 0.019)

3. Body image (p = 0.039)

4. Symptoms experienced (p = 0.043)

Conclusions are presented in an appropriate fashion and are supported by the data.

I would like to thank for the opportunity to review this article and I wish the Authors good luck.

Comment for the editors: I propose to accept the article for publication after taking into account the comments of the reviewer (without the need to re-review).

**Reviewer #2**: The title “Health-related quality of life among cervical cancer survivors at a tertiary hospital in

Ghana” aimed to assess predictors of the HRQoL in cervical cancer survivors. Following comments can be considered in this study.

Minor consideration:

1- Introduction and methods are not well-organized! there are much less-relevant information in methods.

2- Definition of survivors is required. Who are survivors in your study? Are there both short-term and long-term survivors in your study sample?

3- It is multivariable logistic regression (not multivariate)

4- I failed to find any definition of COR and VOR in your study. What are COR and AOR?

5- Use of stars in your results is a little bit non relevant. As mostly one star is used to show significancy at 0.05 level and ** at 0.01 level.

Major consideration:

1- This study has presented in a way that your sample size is representative of survivors in Kumasi, Ghana! However, it seems that there is a lack of external validity in your study. The sample of this study is not a representative sample of cervical cancer survivors in Ghana, while you covering this in your paper. Thus, you cannot generalize your results.

2- On top of that consider the fact that your design is cross-sectional study and prediction is not more accurate in this design. Therefore, you need to take more cautions when interpreting your results and concluding.

3- There is an explanation of adequate sample size but not supported well in the text. As a role of thumb, you need more sample size to conduct your study based on the number of predictors and variables that you entered in your model. I checked the study you referred (reference #27) to check the effect size, but that study was not available. It seems that 3.5 difference in scores is relatively low (assuming your effect size value=3.5). Also, there is small number of samples in some categories such as type of treatments.

4- The outcome is HRQoL! Then you need to determine your sample size based on the poor or good qualified (for instance) that you defined it as poor QOL if scores<0.25 and good score>0.75. then you need to find out the prevalence (using sample size formula based on proportion).

5- Is the cut point of <0.25 (and >0.75) supported in the literature?

6- Results are repeated in discussion! Conclusion is less relevant to your study. Consider the fact that your design is cross sectional. Also, concluding on survival is completely irrelevant.

7- Multivariable logistic regression is not clear. What are your predictors here.? Are you using outcomes item (QoL items) as predictors (e.g., pain, diarrhea) and outcome (Global QoL) of your model!

6. PLOS authors have the option to publish the peer review history of their article (what does this mean?). If published, this will include your full peer review and any attached files.

Reviewer #2: No

---

## [Author Response · Author response to Decision Letter 0]

22 Mar 2022

Department of Obstetrics and Gynaecology, 

Komfo Anokye Teaching Hospital, 

P.O. Box KS 1934, 

Kumasi, Ghana. 

Email : kwabena.amo-antwi@knust.edu.gh
amoantwikwabena@yahoo.com

March 17, 2022

Editor-in-Chief

PLOS ONE

Dear Sir, 

We are grateful for allowing us to submit a revised manuscript titled ‘‘Health-related quality of life among cervical cancer survivors at a tertiary hospital in Ghana.”

We appreciate the time and effort dedicated to providing feedback on our manuscript and the insightful comments and suggestions given to help improve the quality of the manuscript.

We have incorporated the suggestions and answered the queries raised by the reviewers, especially on the methods and materials section. These changes and additional text have been highlighted in blue, and the corresponding lines and page numbers are attached to each query.

Please see below a point-by-point response to the reviewers’ comments or considerations.

Kind regards 

Kwabena Amo-Antwi FWACS FGCS

Corresponding Author

 

Additional Editor comments:

Query: Ethical issue: verbal consent of the participants is a major concern. Do you have any evidence to show that you have gotten the content from the participants? 

Authors response to Editor: Thank you immensely for your concern. Given that most interviews were conducted over the phone, we discussed this and agreed that the most practical way of obtaining informed consent was the verbal consent. The research protocol including the verbal consent procedures was approved by the Committee on Human Research, Publications and Ethics (CHRPE), Kwame Nkrumah University of Science and Technology and KATH (CHRPE/AP/661/19). A study by Karasik et al (PLoS ONE, 2018: 13(10): e0204428) used a similar approach. We have attached a copy of the patient information sheet and consent form. Please refer to lines # 266-274, page 12

Reviewers' Comments to the Authors: 

Reviewer 1 

1. Query: Recommend the use of Dunn's pairwise comparison to elicit differences in the scores of overall QOL (GHS), financial impact, body image, and symptom experience regarding primary treatment received by the survivors.

Authors’ response: Thank you very much for your recommendation. We have included Dunn's pairwise comparison, which further confirmed a difference in the score for overall QOL (p = 0.0036), financial impact (p = 0.0013), body image (p = 0.0013), and symptom experience (p = 0.0068) regarding surgery and chemoradiation (table 5). However, women who had surgery differed from radiation-alone treatment based on body image (p = 0.0212) and financial impact (p = 0.0206). The output for the Dunn's pairwise comparison is provided by table 5. Please refer to Line #363-368, page 20

 

Reviewer 2

Minor considerations 

1. Query: The introduction and method sections are not well organized! There is much less-relevant information in the methods and materials section.

Authors’ response: The write up on the method and materials section has been revised and made more relevant to the study. Steps have been taken to increase the reproducibility of the methods and materials section. 

2. Query: Definition of survivors is required. Who are your survivors in your study? Are there both short-term and long-term survivors in your study sample?

Authors’ response: Survivorship has now been defined in the methodology. “Cancer survivorship begins at the moment of diagnosis and continues through the cancer trajectories (reference 24). Women who survived cervical cancer of 5 years and beyond after diagnosis, with or without and treatment, were defined as long term survivors. Short-term survivors were those who had not lived for a period not up to five years after cancer diagnosis, with or without treatment”. Please refer to Line #184 - 187, on page 8

Defining survivorship necessitated the re-categorisation of the follow-up period and taking cognisance of the cancer diagnosis date (instead of the date of treatment completion). There were short- and long-term cervical cancer survivors. Please refer to table 1, Line #290 - 294, on page 13

3. Query: It is multivariable logistic regression (not multivariate)

Authors’ response: We have replaced “multivariate logistic regression” with “multivariable logistic regression”. Please refer to Lines #47, #262, #371 and #393 on Pages 2, 11, 20 and 22, respectively. Thank you. 

 

4. Query: I failed to find any definition of COR and AOR in your study. What are COR and AOR?

Authors’ response: We have defined and stated how crude odds ratio (COR) and adjusted odds ratio (AOR), symbolized as OR* and OR†, respectively, were obtained in the statistical analysis section of the manuscript. Please refer to line # 60 on page 3 and Table 3 on page 21. And again, they were defined below Table 3. Thank you.

5. Query: Use of the stars in your results is a little bit non relevant. As mostly one star (*) is used to show significance at 0.05 level and two stars (**) at 0.01 level. 

Authors’ response: We have removed stars used to notate p-values in the results section. Please refer to table 6, Line #378- 396, on pages 21 and 22.

Major considerations 

1. Query: This study was presented in a way that your sample size is representative of survivors in Kumasi, Ghana. However, it seems there is a lack of external validity in your study. The sample size of this study is not representative sample of cervical cancer survivors in Ghana, while you covering this in your paper. Thus, you cannot generalize your results. 

Authors’ response: Thank you very much, we acknowledged this in the limitations of the study: “The findings from this single tertiary institution cannot be generalized to the population of cervical cancer survivors in Kumasi, Ghana or in other LMICs, due to selection bias”. Please refer to Line #474 - 475, on page 25.

2. Query: On top of that consider that fact that your design is cross-sectional study and prediction is not more accurate in the design, there you need to take more caution when interpreting your results and concluding. 

Authors’ response: Thank you very much. This has also been acknowledged in the limitations of the study: Cancer survivors' health-related quality of life changes over time; hence, a cross-sectional design is limited in its ability to detect the pattern of overall quality of life across the cancer trajectory. Please refer to Line #472 - 474, on pages 25.

3. Query: There is an explanation of adequate sample size but not supported well in the text. As a role of thumb, you need more sample size to conduct your study based on the number of predictors and variables that you entered in your model. I checked the study you referred (reference #27) to check the effect size, but that study was not available. It seems that 3.5 difference in scores is relatively low (assuming your effect size value=3.5). Also, there is small number of samples in some categories such as type of treatments.

Authors’ response: We assumed that our study's overall QOL score was similar to that observed by Khalil et al. (Quality of life in long-term cervical cancer survivors: results from a single institution, Gynecol Oncol Res Pract. 2015; 2:7.). Sample size calculation was based on the standard deviation of the study mean. Desiring a margin of error of 3.5 percentage points from the actual population overall QoL score, an estimated sample size of 148 had adequate power to detect the sample’s overall QoL score. Using the formula below: 

n = (Z2. SD2) / ME2 = (1.97)2. (21.7)2 /3.52 = (3.8416) (470.89)/12.25=147.67. 

[n-sample size, Z=1.96 for 95% confidence interval, ME-margin of error, SD-standard deviation]. 

However, we do agree that the small number of patients in some categories may limit generalizability of the study findings. We have added this to the limitations of the study. Thank you 

4. Query: The outcome is HRQoL! Then you need to determine your sample size based on the poor or good qualified (for instance) that you defined it as poor QOL if scores < 0.25 and good score>0.75. then you need to find out the prevalence (using sample size formula based on proportion).

Authors’ response: The primary outcome measure is the overall QOL. Please the sample size calculation was based on standard deviation, a similar approach was used by Khalil et al. (2015), an open access article. 

 

5. Query: Is the cut point of <0.25 (and >0.75) supported in the literature?

Authors’ response: Yes please. Araya et al (Health Qual Life Outcomes. 2020, 16;18(1):72) also dichotomized EORTC QLQ-C30 and QLQ-CX24 domain scores. Please see reference number 28. Please refer to lines #557-559#, on page 28.

6. Query: Results are repeated in discussion! 

Authors’ response: The discussion section has been revised, the organization improved and findings that were repeated in the discussion deleted. Thank you.

Query: Conclusion is less relevant to your study. Consider the fact that your design is cross sectional. Also, concluding on survival is completely irrelevant.

Authors’ response: The conclusion has been revised to reflect the objective of the study. 

7. Query: Multivariable logistic regression is not clear. What are your predictors here? Are you using outcomes item (QoL items) as predictors (e.g., pain, diarrheoa) and outcome (Global QoL) of your model!

Authors’ response: The dependent variable in the model overall QOL (GHS) score, the primary outcome measure. The predictor variables were the other QOL subscales (e.g., pain diarrhoea), sociodemographic, clinical and treatment factors.

---

## [Decision Letter · Decision Letter 1]

10 May 2022

Health-related quality of life among cervical cancer survivors at a tertiary hospital in Ghana

PONE-D-21-32659R1

Dear Dr. Kwabena,

We’re pleased to inform you that your manuscript has been judged scientifically suitable for publication and will be formally accepted for publication once it meets all outstanding technical requirements.

Kind regards,

Alison Wang

Academic Editor

PLOS ONE

---

## [Editor Report · Acceptance letter]

26 May 2022

PONE-D-21-32659R1 

Health-related quality of life among cervical cancer survivors at a tertiary hospital in Ghana 

Dear Dr. Amo-Antwi:

I'm pleased to inform you that your manuscript has been deemed suitable for publication in PLOS ONE. Congratulations! Your manuscript is now with our production department. 

Kind regards, 

on behalf of

Dr. Tao (Alison) Wang 

Academic Editor

PLOS ONE